# Reactogenicity and Humoral Immune Response after Heterologous Vaxzevria/Comirnaty Vaccination in a Group of Individuals Vaccinated in the AOU Policlinic “G. Martino” (Messina, Italy): A Retrospective Cohort Study

**DOI:** 10.3390/vaccines10111803

**Published:** 2022-10-26

**Authors:** Roberto Venuto, Ioselita Giunta, Mario Vaccaro, Vincenza La Fauci, Concetta Ceccio, Francesco Fedele, Antonino Privitera, Federica Denaro, Giuseppe Pantò, Rosaria Cortese, Giovanna Visalli, Smeralda D’Amato, Andrea Squeri, Raffaele Squeri, Cristina Genovese

**Affiliations:** 1Department of Biomedical and Dental Sciences and Morphofunctional Imaging, University of Messina, 98124 Messina, Italy; 2Department of Clinical and Experimental Medicine, Unit of Dermatology, University of Messina, 98124 Messina, Italy; 3Department of Human Patology “G. Barresi”, University of Messina, 98124 Messina, Italy

**Keywords:** COVID-19, Comirnaty, Vaxzevria, heterologous vaccination regimen COVID-19, homologous vaccination regimen COVID-19

## Abstract

On 11 June 2021, the Italian Ministry of Health authorized the heterologous vaccination schedule. The goals of our retrospective study were to (a) evaluate the undesirable effects after the administration of Vaxzevria and Comirnaty vaccines; (b) evaluate the antibody response after 28 days from the administration of the second dose; and (c) compare the antibody responses after the homologous and heterologous vaccination regimens. The undesirable effects were collected using a survey; IgG Spike was quantified using the electrochemiluminescence method; the comparison between the antibody responses was carried out using the sample of a homologous vaccine schedule previously analyzed. Pain at the injection site is the most common undesirable effect after the administration of both vaccines (62.1% after Vaxzevria vs. 82.75% after Comirnaty); swelling at the injection site is more frequent after the administration of Vaxzevria than after the administration of Comirnaty: (15.52% vs. 5.17%); headache is more frequent in women than in men for both the vaccination types (*p* < 0.05); 49.09% of the sample reported IgG Spike ≥ 12,500 U/mL; the antibody titer of the heterologous schedule is higher than that of the homologous vaccination. Our study demonstrated that the undesirable effects after the administration of the second dose are less frequent and less severe than after the administration of the first dose, and that the immunogenicity of the heterologous vaccinations is higher than that of the homologous ones.

## 1. Introduction

Coronavirus disease-19 (COVID-19), caused by severe acquired respiratory syndrome coronavirus-2 (SARS-CoV-2) [1], had and still has a global impact, not only in terms of morbidity but also on the socioeconomic side, requiring an outstanding effort by the international scientific community in order to prevent viral transmission and to reduce severity and lethality in patients.

In Europe, the need to produce a vaccine against COVID-19 caused the EMA to quickly authorize the emergency use of mRNA vaccines, BNT162b2 (brand name Comirnaty, developed by Pfizer/BioNTech) and mRNA-1273 (brand name Spikevax, developed by Moderna) and viral vector vaccines, ChAdOx1 n-Cov-19 (brand name Vaxzevria, developed by AstraZeneca) and Ad26.COV2.S (brand name Jannsen, developed by Johnson & Johnson) [2].

COVID-19 vaccination has favorably changed the course of the SARS-CoV-2 pandemic, lowering severe disease and death rates worldwide. Both vaccine types proved to be effective in preventing disease spread and severe symptomatic cases [3].

The changeable epidemiological landscape needs a vaccine strategy that constantly considers the risk–benefit ratio and with pharmacovigilance given a fundamental role.

The increased risk of thromboembolic events reported shortly after inoculation with the Vaxzevria vaccine, especially in young women, caused the Italian Medicines Agency (AIFA) to stop in a precautionary and temporary way (like other European countries) the administration of the aforementioned vaccine in Italy while awaiting an announcement by the European Medical Agency (EMA) [4].

On the 18 March, the EMA, considering the epidemiological landscape at that time, authorized its use again [5].

In April, the Italian Ministry of Health published a circular (prot. n° 14,358) recommending the preferential use of the aforementioned vaccine in individuals above 60 years of age who were not extremely vulnerable and confirming the possibility to complete the vaccination course using the same vaccine for individuals who received the first dose of it.

On the 11 June 2021, the Italian Ministry of Health, acknowledging the changed epidemiological landscape and the increased availability of mRNA vaccines, also authorized the heterologous vaccination in Italy, which involved completing the vaccination cycle with a second dose of a mRNA vaccine (Comirnaty or Spikevax), for individuals under 60 years old and to be administered 8–12 weeks after the administration of the first dose [6,7].

The heterologous regimens are generally well tolerated [8], but currently the literature data on reactogenicity and human immune response after the vaccination regimen are scarce, especially those relating to the Italian population.

Therefore, the goals of our study were:(a)An evaluation of the reactogenicity of the heterologous COVID-19 vaccination, searching for undesirable effects reported by the vaccinated people after the administration of the first dose of the Vaxzevria vaccine and after the administration of the second dose of the Comirnaty vaccine, as well as the presence of statistically significant associations with age, sex, and comorbidities.(b)An evaluation of the antibody response after 28 days from the administration of the second dose, searching for the presence of statistically significant associations with age, sex, and comorbidities.(c)A comparison of the antibody responses after the homologous and heterologous vaccination regimens.

## 2. Materials and Methods

Our study was carried out from June to September 2021.

Each person we proposed to join this study decided to participate.

The study was approved by the AOU Policlinic “G. Martino” Ethics Committee with Protocol 108-22 on 12 April 2022.

### 2.1. Inclusion Criteria

–Age ≥ 18 years.–Administration of a dose of Vaxzevria vaccine.–Administration of a dose of Comirnaty vaccine for completing the vaccination course in the AOU Policlinic “G. Martino” (Messina, Italy).–Administration of a questionnaire before the administration of the Comirnaty vaccine and a week after the administration of the Comirnaty vaccine.

### 2.2. Evaluation of Reactogenicity

A questionnaire was administered to the sample, both during the pre-vaccination anamnesis and a week after completing the vaccination course, by phone interview. This questionnaire (Table 1), based on the current information on the anti-SARS-CoV-2 vaccines used in this study, Vaxzevria and Comirnaty, was made up of three sections: (a)The first one regarded sociodemographic characteristics (Section 1).(b)The second one concerned the presence of comorbidities (Section 2). We divided the enrolled individuals into non-vulnerable, vulnerable, and extremely vulnerable. This classification was based on a table developed by the Sicilian region, defining as extremely vulnerable the individuals affected by some conditions characterized by pre-existing organ damage or by an immune deficiency, with a particularly elevated risk of developing severe or lethal forms of COVID-19. Vulnerable individuals were considered those with at least a chronic disease not included in the previous category. Finally, non-vulnerable individuals were those with no chronic diseases.(c)The third one concerned the presence of undesirable effects reported within 7 days of vaccine administration (Section 3).

As regards symptoms such as fatigue and myalgia/arthralgia, which are very common in the general population, we only reported new cases. Moreover, we assigned adverse reactions to severe symptoms in accordance with the European classification of adverse events by the European Medicines Agency. If in doubt, we considered the judgement of the healthcare worker that performed the interview.

### 2.3. Evaluation of Humoral Immune Response

We collected the antibody levels of both nucleocapsid antibodies and anti-SARS-CoV-2 Spike antibodies of the study subjects, asking them to give us the results; the quantifications of the antibodies were performed out of our ward but in the same laboratory with the electrochemiluminescent immunoassay (ECLIA) method developed by Roche©, Basel, Switzerland (Elecsys Anti-SARS-CoV-2 S^®^). The cutoff value, as suggested by the manufacturer, for anti-nucleocapsid antibodies was 1 cutoff index (COI), whereas the Ig Spike value was 0.8 U/mL.

In order to compare the antibody responses after the homologous and heterologous vaccination regimens, we used a sample homogeneous across sex and age of a homologous vaccination regimen previously analyzed [9].

### 2.4. Statistical Analysis

All the answers in the questionnaire were collected and summarized in an Excel format. As regards the qualitative data (sex, presence of comorbidities, adverse events), absolute and relative frequency were calculated. The quantitative characteristics (age and serological value) were summarized as mean, maximum, and minimum values, and standard deviation.

For estimating the presence of a statistical difference in the sample involved in the study, we performed univariate analysis (*p* < 0.05). For estimating the presence of the statistical difference between the average of the homologous and heterologous schedules, we performed the Student’s *t*-test (*p* < 0.05).

The statistical analyses were performed using the StataCorp. 2009. *Stata Statistical Software: Release 11*. College Station, TX: StataCorp LP.

To compare the homologous and heterologous regimens, we used the propensity score matching method.

## 3. Results

The study sample comprised 174 individuals: 27.6% men and 72.4% women, aged between 21 and 88 years. The characteristics of the sample are described in Table 2.

Undesirable effects are described in Table 3.

These results show that the adverse reactions are moderate or mild for both vaccines, and the most reported undesirable effect after the administration of both vaccines is pain at the injection site: it was reported by 62.0% of the total sample after Vaxzevria and by 82.76% after Comirnaty. Regarding the classification of comorbidities investigated, we did not detect any statistical differences among the several considered as adverse events (Table 4). There is no sex difference with this undesirable effect (Table 3). Furthermore, this effect is more frequent in under-50s (*p* <0.05) (Table 5).

Swelling at the injection site is more frequent after the administration of Vaxzevria than after Comirnaty: 15.52% of the total sample vs. 5.17%.

Both chills and fever ≥ 38.5 °C are more frequent after the administration of Vaxzevria than after Comirnaty. Fever is more frequent in women than in men after the administration of both vaccines (Table 3); however, there are no statistically significant differences.

Headache, considerably less frequent after the administration of Comirnaty than after Vaxzevria, is more frequent in women than in men for both vaccination types (*p* < 0.05) and in under-50s (*p* < 0.05) (Table 5).

Fatigue is more frequent in under-50s (*p* < 0.05) (Table 5).

Furthermore, allergic reactions are more frequent after the administration of Vaxzevria; however, there are no statistically significant differences by age (Table 5) or sex (Table 3).

### Evaluation of the Humoral Immune Response

All the individuals of the study sample reported values above the cutoff indicated by the manufacturer, Roche^®^ (mean: 8993.28 U/mL ± 4060 SD; min 1029 U/mL—max 12,500 U/mL), for the heterologous and homologous schedules (mean: 1003.64 U/mL ± 937.07 SD; min 0.028 U/mL—max 4025 U/mL) (Figure 1).

We compared these values with these from a sample of the homologous schedule with the propensity score matching method: there is significant difference between the two schedules regarding immunization, with higher immunogenicity for the heterologous schedule (*p* < 0.001).

Student’s *t*-test calculation for the presence of a statistical difference:t = (M1 – M2)/√(s2M1 + s2M2) = 8244.91/√230705.16 = 17.17.

## 4. Discussion

This study evaluated the reactogenicity and immunogenicity of the heterologous vaccination.

### 4.1. Reactogenicity and Humoral Immune Response

As regards the first point from the evaluation of the undesirable effects reported by the vaccinated individuals, we highlighted that pain at the injection site is the most common undesirable effect after the administration of both vaccines; moreover, the frequency of undesirable effects after the administration of the heterologous regimen was also evaluated by other studies that had shown that reactogenicity after the boost vaccination is consistently increased in heterologous versus homologous schedules of Vaxzevria and mRNA vaccines [10]. A study conducted in eight sites across the UK showed that heterologous schedules incorporating Vaxzevria (ChAdOx1 nCoV-19, AstraZeneca) and Comirnaty (BNT162b2, Pfizer–BioNTech) at a 4-week interval are more reactogenic than homologous schedules [11].

Furthermore, we determined that swelling at the injection side is more frequent after Comirnaty than after Vaxzevria in contrast with data derived by the European Medical Agency (EMA) [12].

In our study, it was shown that age correlates with greater reactogenicity, similarly to what was evidenced by a Spanish randomized study in which subjects aged between 18 and 60 years received a dose of Comirnaty 2–3 months after the priming with Vaxzevria; although these mild to moderate symptoms were transient, they should be considered during the implementation of the heterologous program, especially in individuals younger than the participants enrolled in this study, given the reported trend towards greater reactogenicity with decreasing age [13,14]. Moreover, the European Medical Agency (EMA) determined that most cases were among patients in the age range of 18–64 and in females both for Comirnaty and Vaxzevria [12].

We noticed that undesirable effects are more frequent in under-50s. A similar result was obtained by a Spanish randomized study enrolling subjects aged between 18 and 60 years, who received a dose of Comirnaty 2–3 months after the priming with Vaxzevria [13,14]. In particular, we observed in under-50s a higher frequency of fatigue, fever, and sleep disorders. In the international literature, the majority of articles found an association between sleep disorders and COVID-19 onset or as sequelae [15], but only a case report showed the relapse of secondary hypersomnia [16].

Regarding sex (Table 3), we did not notice any difference for pain at the injection site, whereas fever and headache are more frequent in women than in men after the administration of both vaccines, as reported by a previous study [17].

Furthermore, we did not notice any statistical difference between the aforementioned three groups of comorbidities, in contrast to the literature [18].

From the evaluation of the humoral responses after 28 days from the immunization, we observed high antibody titers in all investigated subjects. We did not observe any statistical differences by sex, age, or comorbidities, and antibody titers were similar to other studies [19].

Moreover, we found significant differences between the two schedules of immunization, with higher immunogenicity for the heterologous schedule (*p* < 0.001), similar to another study conducted by Liu et al. in which four vaccination programs were evaluated, noting, however, that the Comirnaty-containing schemes were more immunogenic than the homologous Vaxzevria/Vaxzevria scheme [10].

We found that severe adverse events were rare, in accordance with another study [20]. Moreover, we observed that individuals over 50 years of age, of female gender, and with a history of comorbidities are at higher risk of developing effects post COVID-19 vaccination, similar to other studies [20].

### 4.2. Limitations of the Study

The limitations of this study were those of observational studies, such as the possible presence of confounding and distortion bias, including factors such as sex, age, socioeconomic status, lifestyle (smoking, alcohol, and diet), and working status.

Moreover, T- and B-cell response was not evaluated, and it was not possible to evaluate the mucosal immunogenicity. Furthermore, neutralization assays were not performed.

Data about the immunogenicity of heterologous schedules are available for a wider range of vaccine combinations and dosing regimens. These must be interpreted with caution considering the lack of an established correlation of initial or long-term protection, as well as the confounding of schedule with dosing interval in several observational studies [16].

Moreover, our results could be used to implement vaccine adhesion in hesitant people in order to improve COVID-19 vaccine coverage.

## 5. Conclusions

We noticed that the undesirable effects after the administration of the second dose are less frequent and less severe than after the administration of the first dose. We noted that the humoral response of the heterologous vaccinations is higher than that of the homologous ones.

These findings support flexibility in the implementation of viral vector and mRNA vaccines, subject to supply and logistical considerations, and underscore the importance of obtaining information on other programmers with different first boost intervals, particularly for the vaccines used in low- and middle-income countries.

Furthermore, our study demonstrates that the continuous evaluation of adverse events is of paramount importance; despite the possibility of adverse events, most of all mild severity, immunization is the most important weapon, and the use of heterologous prime-boost COVID-19 vaccine schedules could facilitate mass COVID-19 immunization.

Primary prevention, including promotion programs, vaccination, and the prevention of healthcare-associated infections, remains the most important weapon in the hands of public health authorities [21,22,23,24,25].

## Figures and Tables

**Figure 1 vaccines-10-01803-f001:**
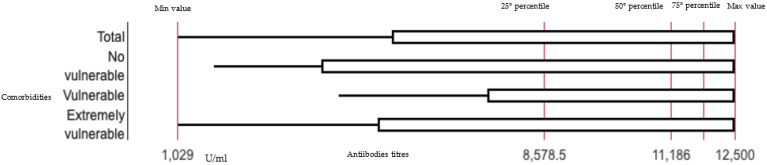
Boxplot representing the distribution of the antibody titers after heterologous schedule on total sample and by comorbidities.

**Table 1 vaccines-10-01803-t001:** Questionnaire.

Section 1. Sociodemographic Characteristics.			
Sex			
Age			
**Section 2. Presence of Comorbidities.**			
		Yes	No
Extremely vulnerable:			
If yes:			
Respiratory diseases			
Cardiocirculatory disease			
Neurological diseases or disability			
Diabetes/other severe endocrinopathies			
Kidney diseases			
Autoimmune diseases			
Liver diseases			
Cerebrovascular diseases			
Onco-hematological diseases or haemoglobinopathies			
Graft			
Severe obesity (BMI > 35)			
Severe underweight (BMI < 16.5)			
Vulnerable:			
If yes, specify what chronic disease: _______________________________________________________			
**Section 3. Presence of Adverse Events**.			
No			
Yes	Specify if a local (i.e., rash, pain, etc.) or general reaction (tiredness, headache, myalgia, arthralgia, chills, fever, etc.), its type, and duration____________________________		

**Table 2 vaccines-10-01803-t002:** Sociodemographic characteristics of the study sample (n = 174).

	N°	%
Sex		
Male	48	27.6
Female	126	72.4
**Mean age ± SD**	51.83 ± 14.40 SD	
**Age**		
21–30	12	6.90
31–39	21	12.07
40–49	39	22.41
50–59	60	34.50
60–69	18	10.35
70–79	23	13.21
>80	1	0.56

**Table 3 vaccines-10-01803-t003:** Undesirable effects reported after the administration of Vaxzevria and after Comirnaty.

	Vaxzevria *		Comirnaty *	
	M (%)	F (%)	M (%)	F (%)
Pain at injection site	44.83% (78)	55.17% (96)	43.10% (75)	56.90% (99)
Fatigue	44.83% (78)	55.17% (96)	18.97% (33)	81.03% (141)
Fever (≥38.5 °C)	46.55% (81)	53.45% (93)	8.62% (15)	91.38% (159)
Chills	29.31% (51)	70.69% (123)	25.86% (45)	74.14% (129)
Headache	48.28% (84)	51.72% (90)	12.07% (21)	87.93% (153)
Nausea	20.69% (36)	79.31% (138)	27.59% (48)	72.41% (126)
Myalgia/arthralgia	34.48% (60)	65.52% (114)	3.45% (6)	96.55% (168)
Vomiting	6.90% (12)	93.10% (162)	3.45% (6)	96.55% (168)
Swollen lymph nodes	3.45% (6)	96.55% (168)	3.45% (6)	96.55% (168)
Diarrhea	3.45% (21)	96.55% (153)	3.45% (6)	96.55% (168)
Swelling at injection site	12.07% (18)	87.93% (156)	1.72% (3)	98.28% (171)
Allergic reactions	10.34% (33)	89.66% (141)	10.34% (18)	89.66% (156)
Sleep disorders	18.97% (12)	81.03% (162)	8.62% (15)	91.38% (159)
Neurological disorders	60.34% (105)	39.66% (69)	0.00% (0)	0.00% (0)

* Percentage is calculated on total sample. Several undesirable effects can coexist in the same person.

**Table 4 vaccines-10-01803-t004:** Adverse events by comorbidities for Vaxzevria and Comirnaty.

Vaxzevria			
Comorbidities	Non-Vulnerable	Vulnerable	Extremely Vulnerable
Pain at injection site	32.76% (57)	12.07% (21)	17.24% (30)
Fatigue	31.03% (54)	12.07% (21)	17.24% (30)
Fever (≥ 38.5 °C)	31.03% (54)	10.34% (18)	17.24% (30)
Chills	22.41% (39)	8.62% (15)	10.34% (18)
Headache	31.03% (54)	10.34% (18)	17.24% (30)
Nausea	10.34% (18)	6.90% (12)	8.62% (15)
Myalgia/arthralgia	18.97% (33)	10.34% (18)	17.24% (30)
Vomiting	3.45% (6)	0.00% (0)	5.17% (9)
Swollen lymph nodes	1.72% (3)	0.00% (0)	1.72% (3)
Diarrhea	1.72% (3)	0.00% (0)	1.72% (3)
Swelling at injection site	3.45% (6)	1.72% (3)	10.34% (18)
Allergic reactions	10.34% (18)	3.45% (6)	1.72% (3)
Sleep disorders	12.07% (21)	5.17% (9)	10.34% (18)
Neurological disorders	5.17% (9)	5.17% (9)	3.45% (6)
**Comirnaty**			
Pain at injection site	39.66% (69)	20.69% (36)	22.41% (39)
Fatigue	29.31% (51)	10.34% (18)	12.07% (21)
Fever (≥ 38.5 °C)	13.79% (24)	3.45% (6)	3.45% (6)
Chills	3.45% (6)	1.72% (3)	3.45% (6)
Headache	13.79% (24)	5.17% (9)	8.62% (15)
Nausea	8.62% (15)	0.00% (0)	3.45% (6)
Myalgia/arthralgia	15.52% (27)	3.45% (6)	12.07% (21)
Vomiting	3.45% (6)	0.00% (0)	0.00% (0)
Swollen lymph nodes	5.17% (9)	0.00% (0)	0.00% (0)
Diarrhea	5.17% (9)	1.72% (3)	1.72% (3)
Swelling at injection site	5.17% (9)	1.72% (3)	1.72% (3)
Allergic reactions	0.00% (0)	1.72% (3)	3.45%
Sleep disorders	0.00% (0)	0.00% (0)	1.72% (3)
Neurological disorders	6.90% (12)	0.00% (0)	5.17% (9)

**Table 5 vaccines-10-01803-t005:** Undesirable effects after both vaccines stratified by age.

Age	<50	51–60	>60
Vaxzevria			
Pain at injection site	28% (48)	26% (45)	9% (15)
Fatigue	29% (51)	17% (30)	14% (24)
Fever (≥38.5 °C)	34% (60)	16% (27)	9% (15)
Chills	22% (39)	16% (27)	3% (6)
Headache	31% (54)	22% (39)	5% (9)
Nausea	17% (30)	7% (12)	2% (3)
Myalgia/arthralgia	24% (42)	12% (21)	10% (18)
Vomiting	7% (12)	0% (0)	2% (3)
Swollen lymph nodes	2% (3)	2% (3)	0% (0)
Diarrhea	0% (0)	0% (0)	3% (6)
Swelling at injection site	7% (12)	7% (12)	2% (3)
Allergic reactions	9% (15)	2% (3)	5% (9)
Sleep disorders	16% (27)	7% (12)	5% (9)
Neurological disorders	3% (6)	9% (15)	2% (3)
**Comirnaty**			
Pain at injection site	40% (69)	26% (45)	17% (30)
Fatigue	31% (54)	16% (27)	5% (9)
Fever (≥38.5 °C)	14% (27)	5% (9)	2% (3)
Chills	2% (3)	7% (12)	0% (0)
Headache	16% (27)	10% (18)	2% (3)
Nausea	5% (9)	7% (12)	0% (0)
Myalgia/arthralgia	14% (24)	12% (21)	5% (9)
Vomiting	0% (0)	2% (3)	2% (3)
Swollen lymph nodes	2% (3)	3% (6)	0% (0)
Diarrhea	3% (6)	3% (6)	2% (3)
Swelling at injection site	2% (3)	0% (0)	3% (6)
Allergic reactions	0% (0)	0% (0)	2% (3)
Sleep disorders	9% (15)	2% (3)	2% (3)
Neurological disorders	9% (15)	3% (6)	2% (3)

## Data Availability

Not applicable.

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
