# Peer review of "Reactogenicity and Humoral Immune Response after Heterologous Vaxzevria/Comirnaty Vaccination in a Group of Individuals Vaccinated in the AOU Policlinic “G. Martino” (Messina, Italy): A Retrospective Cohort Study"

_vaccines, 2022, doi:10.3390/vaccines10111803_

Round 1

Reviewer 1 Report

After reading this manuscript I found what I see as several glaring problems.   Conclusion states the following: "Our study confirms that the heterologous and homologous programs of ChAd and BNT 261 can induce robust immune responses with a first boost interval of 4 weeks"   This is vague and not  supported by the paper.  For example, there is no consideration of B-cells or T-cells (they say as much), nor is there a discussion of about neutralization assays.  In what sense is the immune response even meaningful?  Is there any SI?  As far as I can tell there is no data to support this claim.     From what I know about what is considered standard in this community, it is a problem that the symptoms / questions asked during phone interviews are not given.  Is there any sense of a control population here?  Myalgia / arthralgia is, for example, fairly common.  What was the inherent prevalence of that condition in the population before vaccination?  In other words, to what degree can the authors account for the reported side effects as being due solely to the vaccine?  Are the side effects due, for example, to the active ingredients or the inactive ingredients?  How were the survey results used to identify when a person had a condition associated with the vaccine.   Basically, there is either insufficient information to judge this work or there are fundamental scientific flaws in the work.  Not an `exclusive OR'.   

Author Response

Dear reviwer we send to you the point by point response. It is attached below:

  1. After reading this manuscript I found what I see as several glaring problems.   Conclusion states the following: "Our study confirms that the heterologous and homologous programs of ChAd and BNT 261 can induce robust immune responses with a first boost interval of 4 weeks"   This is vague and not  supported by the paper.  For example, there is no consideration of B-cells or T-cells (they say as much), nor is there a discussion of about neutralization assays.  In what sense is the immune response even meaningful?  Is there any SI?  As far as I can tell there is no data to support this claim.  

We changed the state with :In our study the undesirable effects after the administration of the second dose are less frequent and less severe than after the administration of the first dose. We took note that humoral response of heterologous vaccinations for some association of vaccines is higher than homologous one.

  1.   From what I know about what is considered standard in this community, it is a problem that the symptoms / questions asked during phone interviews are not given.  Is there any sense of a control population here?  Myalgia / arthralgia is, for example, fairly common.  What was the inherent prevalence of that condition in the population before vaccination?  In other words, to what degree can the authors account for the reported side effects as being due solely to the vaccine?  Are the side effects due, for example, to the active ingredients or the inactive ingredients? 

We changed conclusion:

In our study the undesirable effects after the administration of the second dose are less frequent and less severe than after the administration of the first dose.

We add this: Besides, T and B cell response was not evaluated, and it was not possible to evaluate the mucosal immunogenicity. Furthermore, neutralization assays were not performed.

We add this:

As regards symptoms such as fatigue, myalgia/arthralgia, which are very common in the general population, we only reported those ones of new onset.

We remain available for further questions and we attend your response.

Best regards

Reviewer 2 Report

Dear authors,

thank you for the opportunity to read this article

It is really very clear, smooth and pleasant to read.

I believe that the introduction, material, methods and results are very clear.

I think the discussion deserves a more in-depth discussion on the impact of your results and on the perception of the population and health personnel subjected to vaccination.

You have done well to point out the limitations

Author Response

#Reviewer 2

Dear authors,

thank you for the opportunity to read this article

It is really very clear, smooth and pleasant to read.

I believe that the introduction, material, methods and results are very clear.

I think the discussion deserves a more in-depth discussion on the impact of your results and on the

perception of the population and health personnel subjected to vaccination.

You have done well to point out the limitations.

Dear reviewer, thanks for your advice.

So at the end of the discussion, we added: Moreover, our results could be used to implement vaccine adhesion in hesitant people in order to improve COVID-19 vaccine coverage.

Reviewer 3 Report

In this study the authors evaluate antibody responses and reactions to a heterologous vaccine schedule in Italy.  The methodology is fine but I would highly recommend showing the antibody titres for each individual on a graph.
